# Lung Damage Caused by Heated Tobacco Products and Electronic Nicotine Delivery Systems: A Systematic Review

**DOI:** 10.3390/ijerph18084079

**Published:** 2021-04-13

**Authors:** Omar Andrés Bravo-Gutiérrez, Ramcés Falfán-Valencia, Alejandra Ramírez-Venegas, Raúl H. Sansores, Guadalupe Ponciano-Rodríguez, Gloria Pérez-Rubio

**Affiliations:** 1HLA Laboratory, Instituto Nacional de Enfermedades Respiratorias Ismael Cosío Villegas, Mexico City 14080, Mexico; a.bravo.gtz@gmail.com (O.A.B.-G.); rfalfanv@iner.gob.mx (R.F.-V.); 2Tobacco Smoking and COPD Research Department, Instituto Nacional de Enfermedades Respiratorias Ismael Cosío Villegas, Mexico City 14080, Mexico; aleravas@hotmail.com; 3Clínica de Enfermedades Respiratorias, Fundación Médica Sur, Mexico City 14080, Mexico; raulsansores@yahoo.com.mx; 4Public Health Department, Faculty of Medicine, National Autonomous University of Mexico, Mexico City 04510, Mexico; ponciano@unam.mx

**Keywords:** HTP, ENDS, electronic cigarette, inflammation, chronic respiratory disease, DNA damage, oxidative stress

## Abstract

The tobacco industry promotes electronic nicotine delivery systems (ENDS) and heated tobacco products (HTP) as a safer alternative to conventional cigarettes with misleading marketing sustained by studies with conflict of interest. As a result, these devices sell without regulations and warnings about their adverse effects on health, with a growing user base targeting young people. This systematic review aimed to describe the adverse effects on the respiratory system in consumers of these devices. We conducted a systematic review and bibliometric analysis of 79 studies without conflict of interest evaluating ENDS and HTP effects in the respiratory system in experimental models, retrieved from the PubMed database. We found that the damage produced by using these devices is involved in pathways related to pulmonary diseases, involving mechanisms previously reported in conventional cigarettes as well as new mechanisms particular to these devices, which challenges that the tobacco industry’s claims. The present study provides significant evidence to suggest that these devices are an emerging public health problem and that they should be regulated or avoided.

## 1. Introduction

Smoking is a risk factor for developing several chronic diseases (asthma, cancer, chronic obstructive pulmonary disease (COPD)), and it is thus considered a health problem; stricter laws to regulate its consumption have been developed [1,2]. The tobacco industry created Electronic Nicotine Delivery Systems (ENDS), also known as electronic cigarettes (e-cigarettes). They consist of a mouthpiece, a refillable cartridge for heating a liquid matrix or e-liquid (with or without nicotine and other substances), a lithium battery, and a heating atomizer. The users activate the atomizer’s heating coil by depressing the device’s power button during inhalation [3]. More recent generations of these devices allow users to modulate the liquid vaporization process by selecting atomizers with different coil resistances, applying different voltages across the coils, or controlling the atomizer’s operating temperature [4].

The World Health Organization (WHO) has stated that these products are harmful to health and are a gateway for non-smokers to nicotine addiction, especially in young people [5,6]. As for current smokers, the WHO does not recommend these devices as a nicotine replacement therapy [3,7]. In March 2019, the Centers for Disease Control and Prevention (CDC) in the USA reported patient hospitalization as a result of an e-cigarette or vaping-associated lung injury (EVALI). As of January 2020, 37% of these patients were between 18 and 24 years of age [8].

Mexico has been part of the WHO tobacco control initiative since 2003 [9]. In 2008, the General Law for Tobacco Control in Article, 16 Fraction VI, states that it is forbidden to promote and commercialize any object that is not a tobacco product containing any brand or design that associates it with tobacco products [10]; this includes ENDS. Despite this, the national health survey reported in Mexico that the prevalence of e-cigarette consumption was 1.5% (335,100 current ENDS users) in adolescents (10 to 19 years) and 1.2% (1,023,000 current ENDS users) in adults (20 years or more) [11].

In 2014, the tobacco industry presented a new device—Heated Tobacco Products (HTPs)—which heat tobacco and glycerin sticks with low temperatures (less than 350 °C) and without tobacco combustion [3,12]. The WHO statement on these devices is the same as for e-cigarettes [3,13].

The commercialization and promotion of HTPs started in Mexico in September 2019 [14,15]. In February 2020, Mexico forbid the commercialization of e-liquids and ENDS [16]; however, in November 2020, the Supreme Court of Justice of the Nation allowed the marketing of HTPs [17]. The present review’s objective is to describe the findings of studies assessing the safety of these devices and their relationship to human lung damage.

## 2. Materials and Methods

A literature search was performed using the “PubMed” [18] database; the following Medical Subject Heading (MeSH) terms were used: (“electronic * nicotine * delivery * system *” OR “electronic * cigarette *” OR “Heated-tobacco-product *”) AND (“inflammation *” OR “oxidative stress *” OR “DNA-damage *”). Studies in all languages, published between January 2013 and August 2020, were included. The selection criteria included original articles that evaluated ENDS and HTPs and DNA damage, inflammation mechanisms, or reactive oxygen species presence (ROS). The article selection process followed the PRISMA guidelines (Preferred Reporting Items for Systematic reviews and Meta-Analyses) [19]. We performed a factorial bibliometric analysis with the Bibliometrix [20] package in RStudio V 1.4.1103 [21] and determined hierarchical categories for the creation of a proximity matrix using multiple correspondence analysis (MCA) following the workflow proposed by the library’s developer [20].

## 3. Results

We performed a search of papers with the MeSH terms previously mentioned. The search retrieved 222 papers according to the selection criteria; 142 articles were excluded because they were not original research articles (letters to the editor, commentaries, reviews), they were duplicates in the database, they did not report potential conflict of interest, or they had not disclosed results or methodology in the article or supplementary materials (Figure 1). It should be mentioned that studies on HTPs were scarce.

The bibliometric analysis found three main hierarchical topics. We grouped the keywords into three clusters and ran the MCA analysis with 20 words across all articles available to search for relationships in over 80% of the literature reviewed (Figure 2). The conceptual structure analysis using MCA (Figure 3) showed that most reports on ENDS and HTPs are grouped into three categories: in vitro analysis, which includes terms such as cell-line and epithelial cells; animal models, which includes terms such as mice and lung; and human studies, including males, females, and young adults. We utilize these categories to group the findings in the revised articles.

### 3.1. In Vitro Models

Most studies in vitro utilize immortal cell lines from different parts of the lung (bronchial epithelium, lung fibroblast); this is a widely used model because of its low cost and convenience. However, this model sometimes provides varying results, as some of the cellular pathways involved in immortalization tend to interfere with biomarkers released in inflammation and DNA damage [22,23,24]. More recent studies have opted for culturing human bronchial-epithelial (HBE) cells obtained from healthy volunteers.

These cells are exposed to ENDS or HTPs vaporizations or directly to e-liquids in a culture medium [12,25,26,27,28]. Crucial factors such as the cell model employed and the method of vaporization delivery determine the physiological significance of any in vitro study; therefore, more recent studies prefer air–liquid interfaces (ALI) and undiluted aerosols, both of which provide a more pertinent approach for toxicological studies related to inhalation of ENDS and HTP [12,29,30].

In 2014, the Cooperation Centre for Scientific Research Relative to Tobacco (CORESTA) E-Cigarette Task Force (TF) presented standardized parameters for the use of cigarette-machine puffing. These parameters served as a recommended regime for aerosol collection for in vitro studies [31]. However, standardization methodology for assessing HTP emissions seems limited by conventional smoking machines’ capabilities in standard configuration, products of unconventional design, and combinations of volume and puff duration. These recommendations did not consider other factors that have proven to be determinant in assessing the damage dealt by these devices, such as e-cigarette flavors [23,32].

Currently, there are over 15,000 different e-liquid flavors on the market [33]. The Flavor and Extract Manufacturers Association (FEMA) has identified over 1000 flavorings commonly used in e-liquids that may pose a respiratory hazard due to possible volatility and irritant properties. Most studies have identified that aliphatic aldehydes (in fruity flavors), aromatic aldehydes (in sweet and spicy flavors), and non-phenolic terpenes (floral and citric flavors) generate more lung damage [34,35,36]. Another study identified two cinnamaldehyde flavor compounds, ethyl maltol, maltol, and propylene glycol, found in the flavors, as potentially genotoxic [33]. E-liquid without nicotine produced high levels of carbonyl [5].

#### 3.1.1. Cytotoxicity in in vitro Models

The composition of e-liquids changes with the boiling temperature and with the concentration of vegetable glycerin (VG) [37]; the cytotoxic effect is not dependent on formula, brand, or nicotine presence [38,39,40]. E-liquids that are sweet, fruity, and citrus-flavored, as compared to vanilla-flavored or non-flavored, generate more reactive oxygen species (ROS) [36,41]; their presence can initiate pathological processes, oxidative stress, damage of biomolecules (as DNA and protein alteration), and pro-inflammatory responses involved in smoking-related diseases [36].

Cytotoxicity occurs in e-cigarette exposure, assessed by the presence of lactic acid dehydrogenase (LDH). This cytosolic enzyme releases upon damage to the plasma membrane; it has been identified in the supernatant of bronchial epithelial cells (BECs) of healthy non-smokers, COPD patients [23], and immortalized cell-lines (Calu-3 cells) exposed to e-liquid [38,42]. This release is independent of nicotine concentration in alveolar macrophages [43]. Other effects related to cytotoxicity include decreased cell viability in normal epithelial cells and head and neck squamous cell carcinoma cell-lines (HaCats, HN30, and UMSCC10B) [44], induction of apoptosis, mitochondrial dysfunction in human alveolar type II cells (ATII) [45], and autophagy in human embryonic kidney cells (HEK293T) [46]. E-cigarette vapor contains polycyclic aromatic carbonyls, aldehydes [39], and dihydroxyacetone phosphate, among the chemical substances suspected of this damage; these products originate from the solvents’ combustion [46].

The role of nicotine is unclear; some reports have found a significant inflammatory or cytotoxic response to e-liquids containing nicotine. Vaporization of nicotine has proven to generate tobacco-specific nitrosamines, carcinogenic substances, or reactive irritants [47,48]. Nicotine reacts with hypochlorous acid (HOCl), secreted by activated neutrophils, and produces nicotine chloramine (Nic-Cl). This substance produces intracellular protein damage and contributes to chronic inflammation [49]. Nicotine is a chemotactic factor for neutrophils; the alkaloid induces neutrophil extracellular traps (NETs) that contribute to tissue damage and excessive inflammation [50].

In aerosol generated by e-cigarettes, in vitro cells present proteostasis/autophagy impairment [51], suppression of the cells’ ability for DNA to repair damage, disruption in ion homeostasis, oxidative toxicity, membrane dysfunction [52], and cell death via a caspase-independent pathway (in contrast to conventional cigarettes that generally use a caspase-dependent pathway) [42]. The flavored e-liquids alter ion calcium (Ca^2+^) homeostasis, dysregulation of Ca^2+^ signaling (which can cause chronic inflammation), endoplasmic reticulum stress, and abnormal cell growth. Flavored e-liquids with ethyl vanillin, ethyl maltol, and vanillin increase this second messenger [28].

#### 3.1.2. Inflammation

Several lung cells lines (A549, HFL1, NCI-H292, HBE, murine, and human primary alveolar type II cells (ATII) and alveolar macrophages) exposed to ENDS and HTPs produced increased levels of pro-inflammatory interleukin (IL) 8 and IL-6 [22,23,25,34,43,53,54,55]. Other reported differences in markers related to inflammatory processes include IL-1α, IL-1β, IL-10, IL-13, C-X-C motif chemokine ligand 1 (CXCL1), growth-regulated protein alpha (GRO-α), CXCL2, [39], CXCL10, and tumor necrosis factor-alpha (TNF-α) [55,56]. Most of these findings include increased levels of these inflammatory biomarkers. Some studies show that these biomarkers’ secretion seems entirely inhibited by an increased cytotoxic effect, inhibiting the pathways in charge from activating the inflammatory response [34,57].

The S100 protein family includes calcium-binding proteins participating in inflammation, host defense, and carcinogenesis. S100 calcium-binding protein A7 (S100A7) and S100 calcium-binding protein A12 (S100A12) belong to a group of danger-associated proteins, which bind to cell surface receptors, like the receptor for advanced glycation end products (RAGE), and induce inflammation. Additionally, S100A12 produces secreted mucin 5AC (MUC5AC) from airway epithelial cells [22], which is related to an increased intracellular mucin production that could predispose e-cigarette users to airway obstruction similar to chronic obstructive pulmonary disease patients [40].

#### 3.1.3. Airway Infections

Inhalation of contaminants increases airway bacterial infection risk; in vitro, exposure to e-cigarette vapor extract increases platelet-activating factor receptor (PAFR) expression and increases pneumococcal adhesion and infection cells [30]. After exposure, human epithelial cells in the ALI reduced antimicrobial activity against *Staphylococcus aureus* (SA) and induced biofilm formation in methicillin-resistant SA [38]. Acute exposure of human alveolar macrophages and neutrophils to various flavored e-liquids reduced phagocytosis of SA [34]; exposure of alveolar macrophages to e-cigarette vapor condensate (ECVC) reduced *Escherichia coli* bioparticle phagocytosis by 41.7%, and exposure to nicotine-free ECVC (nfECVC) reduced phagocytosis by 48.5% [43]. The THP-1 macrophages exposed to e-vapor extract reduced *Mycobacterium tuberculosis’s* phagocytosis [58] and *Haemophilus influenzae* [55]. Macrophages express a range of cell surface receptors to recognize phagocytic targets, including bacteria; e-cigarette vapor also reduces expression of the phagocytosis receptor, scavenger receptor (SR)-A1, and Toll-like-receptor (TLR)-2 [55].

### 3.2. Animal Models

Studies that use animal models (primarily murine models) have focused on pulmonary, cardiovascular, and central nervous system damage. These observations included alterations in pro-inflammatory and inflammatory mechanisms, DNA damage, and impaired repair mechanisms by e-cigarette exposure [59]. This section includes findings on the pulmonary system after exposure to ENDS or HTP in animal models.

#### 3.2.1. Impaired Lung

The exposure to e-cigarette vapor extract was associated with increased virulence and inflammatory potential of common pathogens involved in respiratory infections (*Haemophilus influenzae, Streptococcus pneumoniae, Staphylococcus aureus*, and *Pseudomonas aeruginosa*). In a model of Galleria mellonella larvae, the medium containing *S. aureus* had a significant increase in biofilm formation. The establishment of biofilm is associated with persistence of infection, resistance to antibiotics, and evasion of the host immune system, raising concerns about the use of e-cigarettes in groups susceptible to infections, like COPD patients [53].

Acute exposure (2 h daily for three days) to e-cigarette aerosols containing propylene glycol (PG) with nicotine induce increased inflammatory cell influx (neutrophils and CD8a + T-lymphocytes) and release of pro-inflammatory cytokines in bronchoalveolar lavage fluid (BALF) of mice [54]. Some authors reported that, in BALF of mice exposed to e-cigarette aerosol, levels of IL-6, IL-1α IL-13, IL-4, and IL-5 were increased [60,61], and glutathione levels were reduced [51]. Sub-acute e-cigarette exposure (2 h a day, 5 days a week for 30 days) evaluated in the BALF of mice showed high levels of TNF-α and macrophage inflammatory proteins (MIP)-1β and IL-1β; this impairment of inflammatory response produces extracellular matrix remodeling of the affected tissue [62].

Chronic inhalation of e-cigarette vapor led to diminished circulating levels of matrix metalloproteinase-3 (MMP-3); lower levels of this protein may be evidence of increased risk of carcinogenesis [26]. Alveolar macrophages of mice with chronic inhalation exposure to e-cigarette vapor showed altered intracellular lipid biosynthesis and increased pulmonary surfactant layer, resulting in downregulated innate immunity against viral pathogens by resident macrophages [42,63]. Mice e-cigarette exposure produced enhanced influenza infection susceptibility, increased percent weight loss, and mortality [47].

Lungs of rats exposed to the vapor generated by e-cigarettes for 28 days at a fixed voltage (3.5 V) and two different coils (1.5 Ω and 0.25 Ω) saw increases of ROS (1.5-fold) in 1.5 Ω exposed group and 2-fold increases in 0.25 Ω, both compared to the control. This model showed increases in IL1-β and IL-6 expression in the 0.25 Ω group and higher neutrophil count. Based on these findings, the authors suggest that e-cigarette consumers need to be cautious with low-voltage devices [64]. This model shared some findings with another study with similar characteristics (3.7 V, 2 Ω coil e-cigarette, red fruit flavor e-liquid, 18 mg/dL nicotine, 4 week exposure to e-cigarette vapor in a puff-machine chamber), reporting overproduction of ROS and induction of cytochrome P450 (CYP) A1/2, CYP2B1/2, and CYP3A in a rat lung model [65]. The family of cytochrome P450 is related to the bioactivation of arylamines, dioxins, aromatic amines, and polycyclic aromatic hydrocarbons (PAHs), which might culminate in DNA adducts, redox imbalance, and lipid peroxidation in erythrocyte membranes [64], with mutagenic and carcinogenic activity [64,65].

The lung histology of mice exposed to e-cigarette aerosol displayed a limited focus of infiltration of inflammatory cells observed after acute exposure [42]. Mice exposed to e-cigarette aerosol displayed decreased parenchymal lung function at both functional residual capacity and high trans-respiratory pressure [66].

#### 3.2.2. Lung Disease Associated with Consumption of ENDS

Clear evidence exists for the association between conventional cigarette smoking and lung diseases; however, few studies in animal models evaluate the risk between the consumption of e-cigarettes and lung diseases. The effects of nicotine-containing e-cigarettes on murine model asthma increased Th2 cytokines (IL-4, IL-5, and IL-13), responsible for allergic inflammation [61]. Exposure to e-cigarette aerosol in mice with ovalbumin sensitization presented increased neutrophils, eosinophils, lymphocytes, and IL-13 but a reduced level of TGF-β1 [67]. The BALF of an animal model of inhaled vitamin E acetate and EVALI-like lung injury reported higher albumin levels, leukocytes, and numerous lipid-laden macrophages; the authors proposed that vitamin E acetate was likely a possible cause of EVALI [68].

### 3.3. Human Studies

The study of damage by ENDS and HTP in humans is complicated. These devices have various flavors and additives. ENDS examples in e-liquids include VG, PG, and other unidentified substances. E-cigarette consumers can change coil temperature and voltage; these modalities affect the compounds produced by high temperatures (100–250 °C) [69].

In the HTP system, the tobacco plugs chars; this charring increase when the device is not cleaned between heat sticks. These devices release formaldehyde cyanohydrin at 90 °C. Even though the HTP operates for a limited time, this is a concern, as it is highly toxic at low concentrations [70,71].

#### 3.3.1. Damage in Human Lungs

Inflammation biomarkers in plasma samples of e-cigarette users had high levels of IL-6, IL-8, IL-13, IFNγ, MMP-9, and IL-1β and lower levels of CXCL1, RAGE, and GM-CSF. Biomarkers for vascular function (ICAM-1) and extracellular matrix breakdown (desmosine) were significantly higher in e-cigarette users compared with non-user subjects. Significant increases in levels of IFN-γ, 8-isoprostane, and 8-oxo-dG are present in urine samples of e-cigarette users [72]. In samples of BALF of e-cigarette users, there are increased counts of macrophages, lymphocytes, and pro-inflammatory cytokines (IL8, IL13, and TNF-α), and a positive correlation was shown with PG levels in urine samples [73].

Expression analysis of e-cigarette users shows upregulation of inflammatory cytokines in bronchial epithelial cells and is associated with inflammasome [74]. Proteomic analysis of induced sputum of e-cigarette consumers showed significant increases of MMP-9, myeloperoxidase, and protein-arginine deaminase 4 as well as an elevated concentration of MUC5AC; these results indicate an altered innate defense and lung structure [75]. In the sputum of vapers, increased secretion of proteases, mucins, and NETs contributed to chronic inflammation [28].

Users of ENDS show reduced endothelial function and increased serum markers of inflammation and oxidative stress, particularly NOX2 (a reduced isoform of nicotinamide adenine dinucleotide phosphate); this relates to the development of endothelial dysfunction and progression toward atherosclerosis [76]. Interestingly, in the serum of people passively exposed to e-cigarette smoke, total antioxidant capacity, catalase activity, and glutathione are reduced, similar to passive smoking of conventional cigarettes [48]. In the urine sample of e-cigarette users, there is a positive correlation between cotinine concentration and total metal concentration, specifically zinc. This metal correlates with oxidative DNA damage [77].

#### 3.3.2. Pulmonary Pathology

Clinical consequences of dysregulation in immunity mechanisms are exemplified in a case report of severe tracheomalacia. A 53-year-old male—an exclusive e-cigarette user for seven years—presented frequent bronchitis episodes; the authors argued that exposure to vaping combined with potentially altered immunological defense led to severe respiratory failure [78]. The pro-inflammatory states induced by these devices and the lipid deposits in the tissue could produce clinical presentations like the case of a 45-year-old female e-cigarette consumer. Her initial imaging showed lung nodules imitating metastatic cancer; the patient’s lung biopsy reported an area with multinucleated giant cells suggestive of a foreign body reaction to a lipophilic material. Upon cessation of e-cigarette use, the lung nodules disappeared [79].

In 2019, some reports appeared in the US about patients who had EVALI. Between January 2018 and August 2019, 53 young people in Wisconsin and Illinois presented bilateral pulmonary infiltrates; 80% of patients had smoked e-liquids containing tetrahydrocannabinol (THC) and 17% used nicotine exclusively [80]. The majority of EVALI patients were healthy adolescents and young adults. The pathophysiology is not well understood; however, speculation points to certain substances such as vitamin E acetate in marijuana-containing vape cartridges [81].

Since that time, more than 2800 cases have been reported to the CDC, resulting in at least 68 deaths. Patients with EVALI presented leukocytosis, elevated erythrocyte sedimentation rate, and high C-reactive protein levels in peripheral blood. The BALF showed a predominance of macrophages and neutrophils; chest radiography revealed multifocal, multilobar opacities, variable in extent and distribution, consistent with foci of alveolar consolidation [82], with the presence of vitamin E acetate in some reports [83]. The most frequent clinical presentation included shortness of breath, cough, chest pain, gastrointestinal symptoms such as abdominal pain, nausea, and vomiting, and neurologic symptoms including headache, lethargy, and confusion [84]. Severe neurologic damage was reported in a case of cytotoxic lesion of the splenium of the corpus callosum in a 17-year-old patient, a daily consumer of THC vaping [85]. Benefits of short-term e-cigarette cessation in regular users include improving airway health status [86].

## 4. Discussion

Evaluation of ENDS and HTPs in different models has demonstrated involvement in pathways related to chronic pulmonary diseases [65,71,72]. As described above, these devices share common pathways of damage and impairment with conventional cigarettes; however, they present with newer damage mechanisms related to the additives, flavors, and metal nanoparticles [31,32,33,34].

We suggest that primary-contact physicians and experts in the field should not recommend these devices as nicotine replacement therapy to smokers, especially in patients with chronic respiratory diseases like COPD and asthma. The damage mechanisms described previously could synergize the underlying immune impairing conditions and exacerbate complications or disease progression [62,71].

Most reports investigated molecules associated with inflammation, oxidative stress, pulmonary remodeling, etc. [42,51,53,65], which were previously explored in conventional cigarettes. Prolonged activation of such mediators may lead to cardiovascular and pulmonary diseases; however, each device of ENDS or HTP has different substances and additives that produce other risk substances associated with lipidic disequilibrium [72,83].

The principal limitations in these studies are establishing puffing topography and the nicotine concentration used. Consumers of these devices have the possibility of selecting a flavor, changing the nicotine concentration, choosing time of use, or choosing dual-use (e-cigarette or HTP and conventional cigarette) [79].

We identified that the target for these devices is young adults and teenagers; therefore, it is crucial to investigate and generate clear evidence of these devices’ adverse effects.

## 5. Conclusions

ENDS and HTPs use is involved in damage related to the development of pulmonary diseases. The evidence available so far is significant enough for physicians, researchers, and public health policymakers to address these devices as an emerging public health problem that needs regulation.

## Figures and Tables

**Figure 1 ijerph-18-04079-f001:**
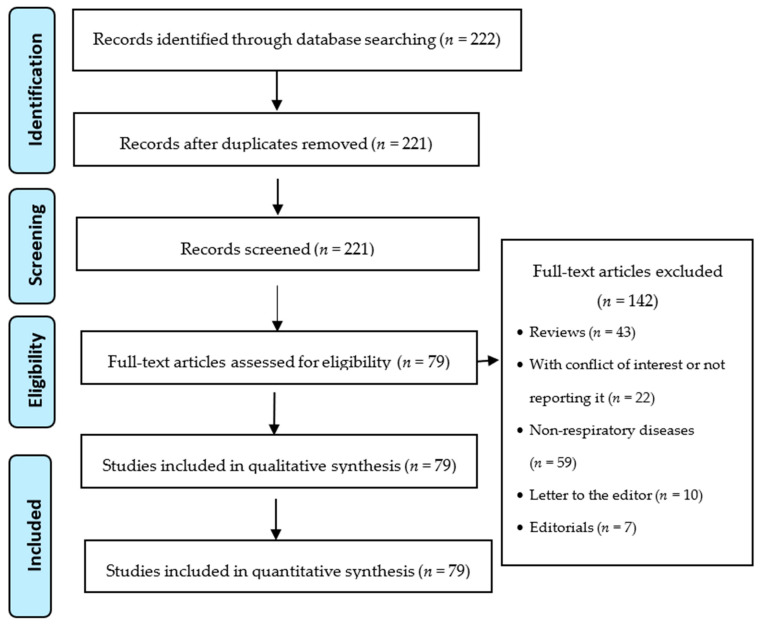
The article selection process followed the PRISMA guidelines (Preferred Reporting Items for Systematic reviews and Meta-Analyses) [19].

**Figure 2 ijerph-18-04079-f002:**
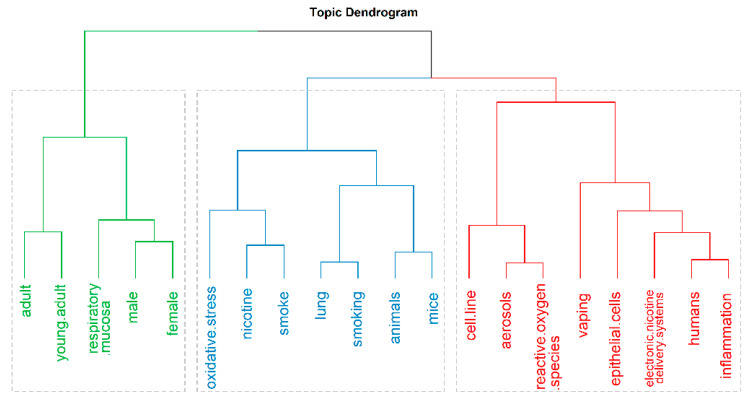
Analysis of the conceptual structure by topic dendrogram determined three hierarchical categories of the keywords included in the articles analyzed. Green = human studies; blue = animal models; red = in vitro studies.

**Figure 3 ijerph-18-04079-f003:**
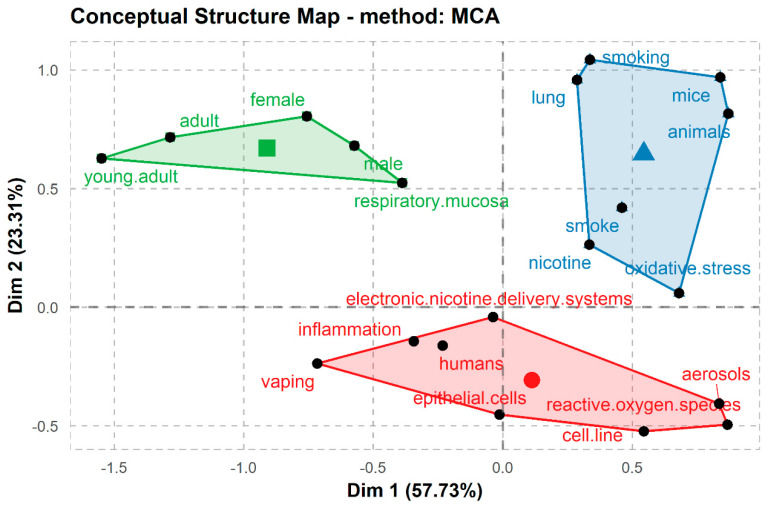
Conceptual structure map obtained by multiple correspondence analysis of the keywords analyzed. Dim = dimension; Green = human studies; blue = animal models; red = in vitro studies.

## Data Availability

No new data were created in this study. Data sharing is not applicable to this article.

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
