# Peer review of "Lung Damage Caused by Heated Tobacco Products and Electronic Nicotine Delivery Systems: A Systematic Review"

_ijerph, 2021, doi:10.3390/ijerph18084079_

Round 1
Reviewer 1 Report
The manuscript of Bravo-Gutierrez and colleagues presents the detrimental effects of heated tobacco products (HTP) and electronic nicotine delivery systems (ENDS) in the lungs under in vitro and in vivo conditions as well as in some patients. The reviewed data indicate that HTP and ENDS consumption mediate lung tissue damage, inflammatory changes in the lungs and susceptibility to pulmonary infections/diseases. Thus, these devices should not be considered as safer alternatives to conventional cigarettes.
This review is systematically presented in a well-prepared manuscript. Yet, some important points will have to be addressed to increase the impact of the review.
Abstract
Lines 28-31, sentence is not clear and has no logical meaning. Authors claim that further studies have to be performed on “users respiratory system in spite of the current studies providing evidence (..) to address these devices as an emerging public health problem”. From my standpoint, the more important message of this project is that, the reviewed literatures have provided evidence that consumption of HTPs and ENDS elicits the same detrimental effects as those of conventional cigarettes. Thus, the usage of these smoking devices are emerging public health problem (lines 367-368) that should be regulated or avoided.
The reviewed studies reported the different lung damages mediated by HTPs and ENDS consumption in the human lungs. It is highly suggested to present these in a graphical summary for a better overview of what have been done and what is to be done to further explore the damaging effects of these smoking devices.
Fig 2 and 3, kindly increase all fonts as these are not readable. Define abbreviations of Dim1 and Dim 2.
Lines 154-156, include a brief functional description of ROS relative to cigarette smoke-induced lung damage.
Lines 162-165, “Additional findings that assess (..) exposed to e-cig with decreased cell viability (44)”. This is intriguing. Does this mean that cytotoxicity mediated by e-cig exposure including apoptosis induction, mitochondrial dysfunction and autophagy were observed in normal epithelial cells and in head and neck squamous carcinoma cells lines? These findings shows that e-cig could exert anti-tumorigenic actions in carcinoma cell lines? A more detailed description of the study is essential.
Line 194, I suggest to use “inflammatory biomarkers” instead of biomarkers.
3.2 Animal models
Lines 229-248, because this review is focused on HTPs and ENDS-induced lung damages, in my opinion, effects in other biological systems are not vital on this occasion.
Line 255, “Subacute exposure (30 days of e-cig exposure). Does this mean 24h treatment per day for 30 days?
Lies 269-271, for clearer understanding, it is suggested to include “based on these findings, authors suggest that e-cig (..) “safer” devices (71)”.
Lines 271-273, “After ling-term exposure (..) and CYP3A (72)”. This is apparently a different study, which was presented as if it was associated with that of ref 71. This should be indicated accordingly.
3.3.1 Damage in lungs humans – kindly change to human lungs
Kindly define abbreviations when first used in the manuscript: CYP1A1/2, COPD, EVALI
Author Response
We appreciate the feedback.

Reviewer 2 Report
Thank you for asking me to review this manuscript. I do have some major concerns about the work and I would like the authors to improve the research.
- The title of the Manuscript: “Lung damage caused by heated tobacco products and electronic nicotine delivery systems: a systematic review” suggests that the results are focused on the electronic cigarettes and Heated Tobacco Products. After an analysis of the Manuscript, I have an impression that the publication is focused only on the e-cigarettes, there is lack of information about HTP. I found information about HTP in the Results part in only 4 places: lines 131, 148-149, 188-189, 299-303. According to that, the study title is confusing, because it says, that we can find information about HTP, and in fact, we cannot. Please, re-write the results part, add more information about HTP or change the title of the Manuscript, as in this form it is not the research about HTP.
- Why terms electronic nicotine delivery systems (ENDS) and heated tobacco products (HTP) are used separately? According to the definition, HTP can be included in the group of ENDS. I understand that as ENDS authors include e-cigarettes but it can be confusing for the reader.
- line 32- Using a term heat-not-burn according to HTP is inappropriate as long as there are evidences that some products of pyrolysis were present during use of HTP, which is even mentioned in this Manuscript. The term H-n-B is used many times across the Manuscript. We should avoid it, especially that the tobacco companies which provide HTP, try to convince people that during the use of HTP there is no combustion.
- line 54: The term “e-cig” sounds colloquially. Isn’t it smoother to use “electronic cigarettes” or “e-cigarettes”? Across the manuscript, the term “e-cig” is repeated many times. Also, please be consequent, if the term “e-cigarettes” is used once, please follow this across the Manuscript. The same with HTP.
- lines 63-66: Please add the reference about the advertisements, where HTPs are promoted as safer than e-cigarettes and conventional cigarettes
- The HTP and e-cigarettes are described interchangeably, sometimes without highlighting, whether the information is about HTP or e-cigarette. It is confusing, for example lines 148-151: “the components between devices” between which devices? cigarettes and HTP? or e-cigarettes and HTP? Also, lines 226-229 – Damages after what? HTP or e-cigarettes? References says that this is about e-cigarettes, but it is not clear from the text. Moreover, Lines 330-334 – I do not know if the vaping is connected here to e-cigarette or to HTP. Please, follow the manuscript and clarify it.
Author Response
We appreciate the feedback.

Round 2
Reviewer 1 Report
Bravo-Gutierrez and colleagues have addressed all the concerns raised the the reviewer accordingly.